# Immunotherapy for Urological Tumors on YouTube^TM^: An Information-Quality Analysis

**DOI:** 10.3390/vaccines11010092

**Published:** 2022-12-30

**Authors:** Francesco Di Bello, Ernesto Di Mauro, Claudia Collà Ruvolo, Massimiliano Creta, Roberto La Rocca, Giuseppe Celentano, Marco Capece, Luigi Napolitano, Agostino Fraia, Gabriele Pezone, Simone Morra, Ciro Imbimbo, Vincenzo Mirone, Nicola Longo, Gianluigi Califano

**Affiliations:** Department of Neurosciences, Reproductive Sciences and Odontostomatology, University of Naples “Federico II”, Via Pansini, 80138 Naples, Italy

**Keywords:** patient nformation, social media, YouTube, urology, urothelial carcinoma

## Abstract

Background: YouTube^TM^ is an open-access source for mass information. Several previous studies of YouTube^TM^ videos showed a high rate of misinformation in the urological field. The aim of the current study was to evaluate the quality of information on immunotherapy (IMT) for urological tumors uploaded to YouTube^TM^. Methods: YouTube^TM^ videos were searched using nine keyword combinations. The PEMAT, the DISCERN tool, and the Misinformation scale were used to assess the quality of information in YouTube^TM^ videos about IMT for urological tumors. Descriptive statistics and Kruskal–Wallis, Chi-square, proportion, and Pearson’s tests were performed. Results: According to the selection criteria, 156 YouTube^TM^ videos were suitable for the analysis and stratified according to topic (urothelial carcinoma vs. renal cell cancer vs. prostate cancer vs. general information on IMT). According to PEMAT A/V, the overall Understandability score was 40% (Inter-Quartile Range [IQR]: 20–61.5) and the overall Actionability score was 0% (IQR: 0–25). According to the DISCERN tool, the overall DISCERN score was 44 (IQR: 39–53.2), defined as “*fair*”. According to the Misinformation scale, we recorded the lowest median overall score for item 4 (“IMT in multimodality approach”) and item 5 (“Future perspective”). Conclusions: YouTube^TM^ cannot be recommended as a reliable source of information on IMT for urological malignancies. In addition, YouTube^TM^ videos contributed to the spread of misinformation by underestimating the role of IMT in a multimodality approach and missing the findings of published clinical trial results.

## 1. Introduction

Immunotherapy (IMT) is changing the way we think about and treat urological malignancies [1]. Particularly, immune-checkpoint inhibitors (ICIs) have revolutionized the management of urological cancers [2,3,4,5,6]. Different signal pathways are involved in immune-checkpoint inhibition, such as Programmed Cell Death Protein 1 (PD-1), Programmed Cell Death Protein Ligand 1 (PD-L1), and anti-cytotoxic T-lymphocyte-associated antigen-4 (CTLA-4) [7]. IMT’s role in the management of urological cancer is strongly corroborated by solid evidence and supported by the most recent European Association of Urology (EAU) guidelines, specifically for renal cell carcinoma (RCC) and for bladder cancer (BC) [8,9,10,11,12,13,14,15,16,17,18,19]. Conversely, for upper tract urothelial carcinoma (UTUC), most of the evidence came from BC trial results, due to the low incidence of the disease [4,20,21,22,23]. Even more uncertain is the role of IMT for prostate cancer (PCa), where the number of patients eligible for IMT management is limited. Regardless of the urological tumor treated, the IMT approach has been previously considered a reasonable option for patients with metastatic disease [21]. Thanks to promising published results, the scientific scenario has changed over the years, including a setting of patients with an earlier stage of disease. For instance, the PURE-01 study assessed the safety of Pembrolizumab in a neoadjuvant setting in muscle invasive bladder cancer (MIBC), obtaining a higher rate of pT0 after surgery relative to previous protocols that did not include IMT [24].

IMT represents a complex topic for people both with and without a medical background. The spread of high-quality information on IMT through social media might represent a potential instrument for education by physicians and by general users who are interested into the topic. Specifically, for patients affected by IMT-eligible diseases, access to more information could increase their awareness in order to make the diagnostic–therapeutic steps more manageable [25,26,27,28,29]. 

The search for and spread of medical information on social media (SoMe) has grown rapidly [30]. Specifically, after COVID-19 outbreak, YouTube™ became one of the most widely used platforms to provide and obtain information [31]. More and more studies have revealed the low quality of medical content uploaded to YouTube™, highlighting the potential risks and consequences of misinformation spreading to Internet users [32,33,34,35]. To the best of our knowledge, we are the first to evaluate the quality of information on IMT in urological tumors, such as urothelial carcinoma (UC), RCC, and PCa, uploaded to YouTube™ during the last decades.

## 2. Materials and Methods

### 2.1. Search Strategy and Video Selection Criteria

On 25 March 2022, from 9:00 a.m. to 9:00 p.m. UTC-4, a YouTube™ systematic search was performed with 9 keyword combinations, examining the first 30 videos for each search (Appendix A). To avoid any research bias, the YouTube^TM^ search was performed in incognito mode. The default YouTube^TM^ search setting does not apply any filter and sorts the results by relevance. A total of 270 videos was obtained. The following exclusion criteria were applied (Figure 1): (i) duplicate videos (*n* = 83), (ii) no information reported on immunotherapy for urological disease (*n* = 18); (iii) length > 50 min (*n* = 12); (iv) non-English language (*n* = 1). For each video, the following videographic characteristics were collected (Table 1): length (seconds), number of views, persistence on YouTube™ (days), number of thumbs-up, number of comments, number of channel subscribers, view ratio (defined as the ratio between number of views and persistence on YouTube™), whether comments were disabled, video author category (defined as a medical association [such as physicians’ community channel, peer-reviewed online journal, and cancer research associations]; medical center, hospital or university; or other [such as television channel and non-profit foundations]), and video topic (defined as UC, RCC, general information on IMT, or PCa). According to recent YouTube^TM^ rules, the number of thumbs-down is no longer available to Internet users. 

### 2.2. Quality and Misinformation Assessment Tools

The quality of videos was assessed by two investigators (a junior and a senior urology resident). A third investigator (an Associate Professor) adjudicated any differences, and a consensus was achieved among all reviewers. The following tools were used to assess the video quality information. First, the Patient Education Materials Assessment Tool for audio-visual content (PEMAT A/V) [36] is a validated method to evaluate and measure the Understandability (the first 13 questions) and Actionability (four questions) of multimedia content (Table 2). Higher results mean that the video content has an educational role for Internet users, who could learn and reproduce the information reported in real life. Specifically, for each score, a good-quality video cut-off must be ≥ 70% [36]. 

Second, the DISCERN questionnaire [35] has been validated to assess the reliability of information sources about treatment choices for health problems (Table 3). The DISCERN tool is composed of Section 1 (question 1–8) evaluating the reliability of sources’ information on the treatment choices, Section 2 (question 9–15) analyzing the details of information reported, and Section 3 (question 16) representing an overall quality rating. For each question, five possible answers are permitted (from 1 = strongly disagree to 5 = strongly agree). The DISCERN final score is categorized as “*excellent*” (range 63–80), “*good*” (range 51–62), “*fair*” (range 39–50), “*poor*” (range 27–38), or “*very poor*” (range 16–26). The assessment of DISCERN was based on the agreement with the EAU clinical practice guidelines on IMT strategies for urological tumors. Thus, higher scores mean that the information reported is high-quality and evidence-based. 

Third, a 5-item Misinformation scale was appositely created for the study, based on the EAU guidelines for IMT strategies for urological tumors (Table 4). It evaluated how accurate the information reported in videos was. The Misinformation scale items consisted of (i) “*Logical sequence of information*”; (ii) “*Therapeutic effects*”; (iii) “*Treatment-emergent adverse effects*”; (iv) “*IMT in multimodality approach*”; (v) “*Future perspective*”. Each item was rated from 1 (meaning high level of misinformation) to 5 (meaning low level of misinformation). Finally, the Misinformation score was calculated as the mean of the 5-item evaluation for each video. Higher results represented more accurate information content.

### 2.3. Statistical Analyses

Descriptive statistics are presented as medians and interquartile ranges (IQR) for continuously coded variables or counts and percentages for categorically coded variables. Kruskal–Wallis, Chi-square, and proportion tests examined the statistical significance in medians’ and proportions’ differences. Pearson’s test was used to assess a potential correlation between the variables. The overall collected videos were stratified into four groups, according to video topic (UC, RCC, PCa, or general information on IMT). In all statistical analyses, the R software (www.rproject.org, accessed on 26 March 2022) environment for statistical computing and graphics (R version 4.0.0, R Development Core Team, Auckland, New Zealand) was used. All tests were two-sided with a level of significance set at *p* < 0.05.

## 3. Results

### 3.1. Videographic Characteristics and Video-Quality Assessment by Topic

Of all 270 videos examined, 156 were suitable for the analysis (Table 1). The videos were stratified by topic: 67 (42.9%) were about UC, 38 (24.3%) about KC, 31 (19.8%) about general information on IMT, and 20 (12.8%) about PCa. 

#### 3.1.1. Urothelial Carcinoma

The median length, number of views, and number of thumbs-up were 342 (IQR: 168.5–1135) seconds, 420 (IQR: 118–2003.5), and 3 (IQR: 1–13.5), respectively. Moreover, across the sample, the median view ratio was 0.4 (IQR: 0.1–3.3). According to PEMAT A/V, the median Understandability score was 40% (IQR: 20–61.5) and the median Actionability score was 0% (IQR: 0–0). According to DISCERN, the median overall score was 45 (IQR: 40-–55.5). Finally, the median Misinformation score was 2.8 (IQR: 2.4–3.6) and the lowest score was recorded for item 5 (2, IQR: 2–3.5). 

#### 3.1.2. Renal Cell Cancer

The median length, number of views, and number of thumbs-up were 166.5 (IQR: 89–408.8) seconds, 369 (IQR: 174.2–896), and 2.5 (IQR: 1–11), respectively. The median view ratio was 0.4 (IQR: 0.1–1.1). According to PEMAT A/V, the median Understandability score was 26.1 % (IQR: 20–44.1) and the median Actionability score was 0% (IQR: 0–18.8). According to DISCERN, the overall median score was 43.5 (IQR: 41–52). Finally, the median Misinformation score was 3 (IQR: 2.2–3.6) and the lowest score was recorded for items 3 and 4 (2, IQR: 2–4 and 2, IQR: 2–3.8, respectively).

#### 3.1.3. Prostate Cancer

The median length, number of views, and number of thumbs-up were 573.5 (IQR: 294–1588.5) seconds, 1570 (IQR: 213.8–4612), and 10.5 (IQR: 0.8–51), respectively. The median view ratio was 1 (IQR: 0.3–6.6). According to PEMAT A/V, the median PEMAT Understandability score was 28.2% (IQR: 20–52.1) and the median PEMAT Actionability score was 0% (IQR: 0–0). According to DISCERN, the median score was 46.5 (IQR: 41–53). Finally, the median Misinformation score was 2.3 (IQR: 2.4–4) and the lowest score was recorded for item 4 (2.5, IQR: 2–3).

#### 3.1.4. General Information on Immunotherapy

The median length, number of views, and number of thumbs-up were 303 (IQR: 210.5–912) seconds, 33242 (IQR: 8247.5–126449), and 261 (IQR: 93–913.5), respectively. The median view ratio was 33.5 (IQR: 4.8–161.5). According to PEMAT A/V, the median PEMAT Understandability score was 61.5 % (IQR: 50–75) and the median Actionability score was 25% (IQR: 0–50). According to DISCERN, the median score was 45 (IQR: 36–49.5). Finally, the median Misinformation score was 2.8 (IQR: 2.4–3.5) and the lowest score was recorded for items 3, 4, and 5 (2, IQR: 2–4; 2, IQR: 1–3; and 2, IQR: 1–3, respectively).

### 3.2. Variable Correlations

A statistically significant positive correlation between length and Understandability (*r* = 0.31, *p* = 0.001), length and Actionability (*r* = 0.17; *p* = 0.02), length and total DISCERN Section 1 (*r* = 0.48, *p* < 0.001), length and total DISCERN Section 2 (*r* = 0.43, *p* < 0.001), length and DISCERN question 16 score (*r* = 0.30, *p* < 0.001), length and total DISCERN score (*r* = 0.50, *p* < 0.001), and length and Misinformation score (*r* = 0.45, *p* < 0.001) was recorded. A statistically significant positive correlation between view ratio and Understandability (*r* = 0.21, *p* = 0.006) and view ratio and Actionability (*r* = 0.21, *p* = 0.01) was recorded. No statistically significant results were achieved for the other correlations (all *p* > 0.05).

## 4. Discussion

The interest in IMT for urological malignancies is exponentially growing due to the intriguing results that are emerging in the current scientific context [3,4,6,37,38,39,40,41,42,43,44]. The U.S. Food and Drug Administration and the European Medicine Agency approved ICIs as standard-of-care across many cancer types [45]. 

The current study aimed to evaluate the quality of the available information on IMT in urological tumors uploaded to YouTube™. We recorded several interesting observations. The majority of YouTube™ videos on IMT were focused on UC and were uploaded by medical associations. Thus, we expected an adequate and exhaustive presentation of information on IMT for UC. However, the PEMAT Understandability and Actionability results were unsatisfying. Concordantly to our results, Capece et al. recorded an Understandability score of 57.8% and an Actionability score of 0% after the evaluation of YouTube^TM^ videos on penile prostheses [46]. Similar observations were recorded in other recent studies [46,47,48]. These findings concerning the IMT field may be explained by the fact that IMT is a complex and specialist topic, challenging to explain and to be understood by the general public. In conclusion, regardless of the topic, YouTube^TM^ videos are not informative enough for applications by Internet users. Thus, future video authors should simplify YouTube^TM^ information on IMT for urological tumors, providing supplementary instruments to make the information uploaded more applicable. Furthermore, according to the DISCERN results, the information provided in YouTube™ videos focusing on UC IMT was “fair”, with an improvable range in publication reliability and treatment choices. The same results were observed in the other topic categories. For example, we recorded an even lower score according to the PEMAT and DISCERN tools for videos focused on RCC. Concerning results were also observed in the Misinformation scale assessment. The lowest value was ranked for the items 4 and 5, defined as “IMT in multimodality approach” and “Future perspective”, regardless of the urological tumor considered. The items were presented with scant information, not representing the current clinical practice and scientific context. It is important to know that several trials on IMT strategies for urological cancer management have been published and others are still on-going, and both patients and even physicians must be informed on the scientific progress achieved [4]. IMT is establishing a new reality in urological cancer management [49,50,51]. Moreover, clinical trials may particularly represent an opportunity for patients who were not eligible for the current standard of care [3,4]. Additionally, IMT can reduce the adverse effects of standard chemotherapy and provide to patients an access to surgical management with a better performance status [24,38,52]. To the best of our knowledge, we are the first to evaluate videos on IMT for urological cancers uploaded to YouTube^TM^. We analyzed as many videos as possible, collecting the first 30 videos for each keyword (nine, for a total of 270 videos). Thus, we made the analysis as representative as possible of current YouTube^TM^ results on the topic. Despite these observations, several further studies are required to better understand and characterize the quality of Internet information on IMT. 

Due to the complexity of the topic and the involved patients, the Internet and particularly YouTube^TM^ may represent a crucial instrument to improve the communication between physicians and patients [53]. Indeed, videos could be an effective tool to improve patient awareness thanks to the visual provision of information, which could overcome some health literacy barriers [54]. Moreover, YouTube^TM^ videos should be uploaded to SoMe platforms to increase the high-quality spread of information. We hope that medical institutions will improve the information provided through YouTube^TM^ videos to enhance an educational role for the platform for Internet users. 

A separate mention must be made of the PCa YouTube™ videos. Specifically, despite the scientific interest in PCa treatment options, only 20 YouTube™ videos concerning IMT for PCa were recorded in the current analysis. The role of IMT in PCa is not deeply studied yet and not attractive for SoMe platforms with respect to surgical, radiotherapy, or medical approaches and their consequences [27,28]. 

Taken together, according to the tools adopted, YouTube^TM^ videos are an unreliable source of information on IMT for urological tumors. The major lacking fields were the outdated information on multimodality approaches including IMT and the future perspectives. Indeed, from the video viewing, it did not emerge that the benefits for patients receiving IMT are significantly higher relative to patients managed with conventional chemotherapy strategies [55]. These observations are applicable for both urothelial tumors, where IMT has an established role, and PCa, where IMT represents a marginal treatment option. These findings lead to an inapplicability of YouTube^TM^ videos to the daily life of both physicians and patients. Consequently, the scientific community must improve online medical content by producing more understandable and reliable materials for Internet users, sharing evidence-based information.

Our study is not devoid of limitations. First, some reliable or non-reliable videos might have been missed, due to our search terms. To mitigate this selection bias, we used nine search keywords combinations. Second, YouTube^TM^ search results rely on Google’s proprietary search algorithms based on users’ previous search activities and location. To minimize this bias, before searching, any personal accounts were logged out and searches were performed in incognito mode. Third, quality assessment of videos were subjective. To reduce this confounder, three investigators were involved to independently analyze video content. However, a multidisciplinary objective analysis with new quality-assessment tools is required in the future [56]. Regardless of these limitations, the present study can be considered a snapshot of the currently available information in YouTube^TM^ videos on IMT for urological tumors.

## 5. Conclusions

YouTube^TM^ is an open-access source for mass information and the overall consideration of IMT for urological tumors in YouTube^TM^ videos is inadequate. Despite the multimedia content being evidence-based, it did not have an educational role for Internet users. Moreover, YouTube^TM^ videos contributed to the spread of misinformation, specifically underestimating the role of IMT in a multimodality approach and omitting the results achieved in current clinical trials. Official medical institutions should improve their multimedia content by producing easily understandable materials for Internet users and sharing evidence-based content. Both health-care workers and the general public must be aware of the potential role of YouTube^TM^ as an educational instrument concerning IMT’s advantages and disadvantages in the treatment of urological tumors.

## Figures and Tables

**Figure 1 vaccines-11-00092-f001:**
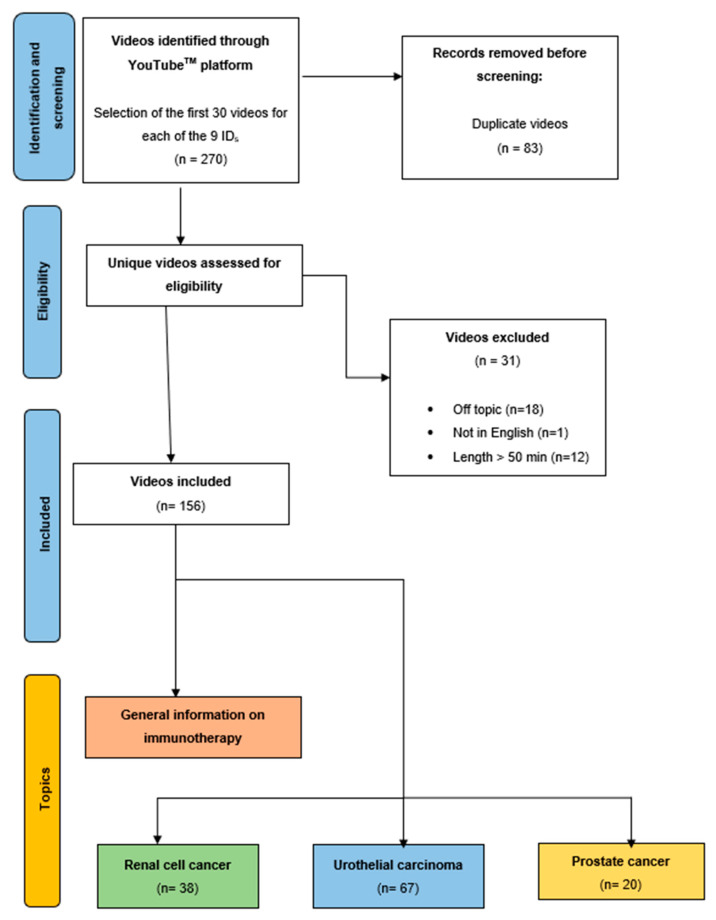
PRISMA diagram depicting inclusion and exclusion criteria of YouTube video search.

**Table 1 vaccines-11-00092-t001:** Videographic characteristics of 156 YouTube videos on immunotherapy in urological tumors found on 25 March 2022, stratified according to the video topic (urothelial carcinoma, renal cell cancer, prostate cancer, or general information on immunotherapy).

Characteristics		Overall N = 156	UC N = 67 (42.9%)	RCC N = 38 (24.3%)	PCa N = 20 (12.8%)	General Information on IMT N = 31 (19.8%)	*p* Value
Length (sec)	Median (IQR)	293.5(151–971.8)	342 (168.5–1135)	166.5 (89–408.8)	573.5 (294–1588.5)	303 (210.5–912)	0.3
Number of views	Median (IQR)	691.5 (224–8041)	420 (118–2003.5)	369 (174.2–896)	1570 (213.8–4612)	33242 (8247.5–126,449)	<0.001
Persistence on YouTube™ (days)	Median (IQR)	1365 (614.5–1795.5)	1269 (560–1695.5)	1608.5 (897.5–1919.5)	1324.5 (622–1635.5)	1359 (637–1751.5)	0.5
Thumbs up (*n*)	Median (IQR)	7 (1–64.2)	3 (1–13.5)	2.5 (1–11)	10.5 (0.8–51)	261 (93–913.5)	<0.001
Number of comments	Median (IQR)	1 (0–6)	0 (0–1)	0 (0–1)	0 (0–6)	17 (5–93)	<0.001
Channel subscribers (*n*)	Median (IQR)	16,900 (4500–30,375)	12,200 (4500–16,900)	16,900 (4590–17,500)	30,600 (17,900–140,100)	41,700 (7300–684,000)	0.07
View ratio	Median (IQR)	1 (0.2–6.2)	0.4 (0.1–3.3)	0.4 (0.1–1.1)	1 (0.3–6.6)	33.5 (4.8–161.5)	<0.001
Videos with disabled comments *n* (%)	No	135 (86.5)	55 (82.1)	34 (89.5)	19 (95)	27 (87.1)	0.2
	Yes	21 (13.5)	12 (17.9)	4 (10.5)	1 (5)	4 (12.9)	
Video author *n* (%)	*Medical Association*	85 (54.5)	38 (56.7)	26 (68.4)	5 (25)	16 (51.6)	0.01
	*Medical Center, Hospital, or University*	35 (22.4)	11 (16.4)	5 (13.2)	9 (45)	10 (32.3)	0.01
	*Other*	36 (23.1)	18 (26.9)	7 (18.4)	6 (30)	5 (16.1)	0.5

“View ratio” is defined as the ratio between number of views and persistence on YouTube (days). Abbreviations: IMT: immunotherapy; IQR: Inter-Quartile Range; PCa: Prostate cancer, RCC: Renal cell cancer; UC: Urothelial carcinoma.

**Table 2 vaccines-11-00092-t002:** PEMAT audio/visual (A/V) scores of 156 YouTube videos on immunotherapy in urological disease, found on 25 March 2022, stratified according to the video topic (urothelial carcinoma, renal cell cancer, prostate cancer, or general information on immunotherapy).

		Overall N = 156	UC N = 67 (42.9%)	RCC N = 38 (24.3%)	PCa N = 20 (12.8%)	General Information on IMT N = 31 (19.8%)	*p* Value
PEMAT Understandability	Median (IQR)	40 (20–61.5)	40 (20–61.5)	26.1 (20–44.1)	28.2 (20–52.1)	61.5 (50–75)	<0.001
PEMAT Actionability	Median (IQR)	0 (0–25)	0 (0–0)	0 (0–18.8)	0 (0–0)	25 (0–50)	<0.001

Abbreviations: IMT: immunotherapy; IQR: Inter-Quartile Range; PCa: Prostate cancer; PEMAT: Patient Educational Assessment Tool; RCC: Renal cell cancer; UC: Urothelial carcinoma.

**Table 3 vaccines-11-00092-t003:** DISCERN instrument Section 1, Section 2, and Section 3 of 156 YouTube videos plus total DISCERN score (a sum of the three sections that could range from a minimum score of 16 to a maximum score of 80), stratified according to the video topic (urothelial carcinoma, renal cell cancer, prostate cancer, and general information on immunotherapy).

			Overall N = 156	UC N = 67 (42.9%)	RCC N = 38 (24.3%)	PCa N = 20 (12.8%)	General information on IMT N = 31 (19.8%)	*p* Value
**Section 1:** **Is the publication reliable?**	**SUM Section 1** **(max score 40)** Question 1–8	Median (IQR)	20 (16–24)	20 (17–24.5)	21 (15.2–23)	20.5 (16–24)	16 (14–20.5)	0.3
**Section 2:** **How good is the quality of information on treatment choices?**	**SUM Section 2** **(max score 35)** Question 9–15	Median (IQR)	20 (16–24)	20 (17–24.5)	21 (15.2–23)	20.5 (16–24)	16 (14–20.5)	0.01
**Section 3: Overall Quality rating**	16. Based on the answers to all the above questions, rate the overall quality of the publication as a source of information about treatment choices **(max score 5)**	Median (IQR)	4 (4–4)	4 (3–4)	4 (4–4)	4 (4–4)	4 (3.5–5)	0.5
	** *DISCERN score* **	Median (IQR)	44 (39–53.2)	45 (40–55.5)	43.5 (41–52)	46.5 (41–53)	45 (36–49.5)	0.2

DISCERN score is categorized as “excellent” (range 63–80), “good” (range 51–62), “fair” (range 39–50), “poor” (range 27–38), or “very poor” (range 16–26). Abbreviations: IMT: immunotherapy; IQR: Inter-Quartile Range; PCa: Prostate cancer, RCC: Renal cell cancer; UC: Urothelial carcinoma.

**Table 4 vaccines-11-00092-t004:** Misinformation scale evaluating 156 YouTube videos on immunotherapy in urological disease, found on 25 March 2022, stratified by video topic (urothelial carcinoma, renal cell cancer, prostate cancer, and general information on immunotherapy).

		Overall N = 156	UCs N = 67 (42.9%)	RCC N = 38 (24.3%)	PCa N = 20 (12.8%)	General information on IMT N = 31 (19.8%)	*p* value
1.“Logical sequence of information”	Median (IQR)	4 (3–4)	3 (3–4)	4 (2–4)	4 (3–4)	4 (3–5)	0.01
2.“Therapeutic effects”	Median (IQR)	4 (3–4)	3 (3–4)	4 (2–4)	4 (3–4)	4 (3–5)	0.3
3.“Treatment-emergent adverse effects”	Median (IQR)	3 (2–4)	4 (2–4)	2 (2–4)	3 (2–4)	2 (2–4)	0.8
4.“IMT in multimodality approach”	Median (IQR)	2 (2–3)	3 (2–3)	2 (2–3.8)	2.5 (2–3)	2 (1–3)	0.2
5.“Future perspective”	Median (IQR)	2 (1.8–4)	2 (2–3.5)	3 (1–4)	3 (2–4)	2 (1–3)	0.03
Misinformation score	Median (IQR)	2.8 (2.4–3.6)	2.8 (2.4–3.6)	3 (2.2–3.6)	2.3 (2.4–4)	2.8 (2.4–3.5)	0.6

A 5-item Misinformation scale was created for this study, based on EAU and ESMO clinical practice guidelines on immunotherapies in urological disease. Each item was rated from 1 to 5 (1—Poor quality; 2—Low quality; 3—Intermediate quality; 4—Good quality; 5—High quality). Quality refers to the amount of information given to YouTube users. Misinformation score is defined as the mean of the 5 items for each video. Abbreviations: IMT: immunotherapy; IQR: Inter-Quartile Range; PCa: Prostate cancer, RCC: Renal cell cancer; UC: Urothelial carcinoma.

## Data Availability

Data available on request due to restrictions eg privacy or ethical.

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
