# Peer review of "Immunotherapy for Urological Tumors on YouTubeTM: An Information-Quality Analysis"

_vaccines, 2022, doi:10.3390/vaccines11010092_

Round 1

Reviewer 1 Report

This is an important contribution to the field of citizen science and public outreach of a common cancer-type. YouTube is also the right platform to assess with its frequency and worldwide reach. I suggest that the author/s need to work on the clarity of the scales and tests performed for general audience. The technical information and data target scientists in the field, but the overall result and findings must be accessible to public and specifically for health care workers with less technical expertise for awareness on the topic.

Author Response

Reviewer 1:

1)  This is an important contribution to the field of citizen science and public outreach of a common cancer-type. YouTube is also the right platform to assess with its frequency and worldwide reach. I suggest that the author/s need to work on the clarity of the scales and tests performed for general audience. The technical information and data target scientists in the field, but the overall result and findings must be accessible to public and specifically for health care workers with less technical expertise for awareness on the topic.

We thank the Reviewer for the comments and suggestions. According to the suggestion of make easily readable the findings reported, we implemented the materials and method, and conclusion sections, which now read as follows:

Materials and methods section:

(p 2-3, line 86-113):” First, Patient Education Materials Assessment Tool for audio-visual content (PEMAT A/V) (38) is a validated method to evaluate and measure the Understandability (the first 13 questions) and Actionability (four questions) of multimedia content (Table 2). PEMAT A/V consists of 17 questions (13 for the Understandability and four for the Actionability of the content).  Higher results means that the video content has an educational role for the Internet users, that could assume and reproduce the information reported into the real life. Specifically, for each score, a good quality video cut-off must be ≥ 70%.

Second, DISCERN questionnaire (37) has been validated to assess the reliability of information source about health problem treatment choices. DISCERN tool is composed of section 1 (question 1-8) evaluating the reliability of sources information on the treatment choices, section 2 (question 9-15) analyzing details of information reported plus section 3 (question 16) representing overall quality rating. For each question, five possible answers are permitted (from 1=strongly disagree, 2=disagree, 3=not agree not disagree, 4=agree, up to 5=strongly agree).  The total sum could range from a minimum score of 16 to a maximum score of 80. DISCERN final score has been categorized as follows: “excellent” (range 63-80), “good” (range 51-62), “fair” (range 39-50), “poor” (range 27-38), “very poor” (range 16-26). The assessment of DISCERN was based on the agreement to the EAU clinical practice guidelines on IMT strategies for urological tumors. Thus, higher results means that the information reported is high quality evidence based.

Third, a 5-item Misinformation scale was appositely created for the study, based on the EAU guidelines referred to IMT strategies for urological tumors. It evaluated how information reported in videos was accurate. The Misinformation scale items consisted of i) “Logical sequence of information”; ii) “Therapeutic effects”; iii) “Treatment-emergent adverse effects”; iv) “IMT in multimodality approach”; v) “Future perspective”. Each item was rated from 1 (meaning high level for misinformation) to 5 (meaning low level of misinformation). Finally, the Misinformation score was calculated as the mean of 5-item evaluation for each video. Higher results stand for more accurate information content.

Conclusion section

In abstract:

 (p 1):” Conclusions: YouTubeTM cannot be recommended as a reliable source of information on IMT for urological malignancies. In addition, YouTubeTM videos contributed to spread misinformation underestimating the role of IMT in a multimodality approach and missing the finding achieved from the clinical trials published results”.

In the main text:

            (p 9-10, lines 309-313): “Despite the multimedia content was evidence based, it had not an educational role for Internet users. Moreover, YouTubeTM videos contributed to spread misinformation specifically underestimating the role of IMT in a multimodality approach and omitting the results achieved from current clinical trials. The official medical institutions should improve the multimedia contents: producing easily understandable materials for Internet users, sharing evidence-based content.  

Reviewer 2 Report

The paper evaluates the quality information on immunotherapy (IMT) for urological tumors uploaded on YouTube. The work is interesting and innovative but can be improved. As suggestions for improvement, I highlight the points below:

The introduction must contain relevant information to contextualize the reader with the theme addressed by the study, citing relevant works that support the points of view presented. In this sense, the article presents several statements without proper citations, which should be corrected to give greater robustness and reliability to the study.

The introduction should contain relevant information to contextualize the reader with the theme addressed by the study and what are its main contributions. In this sense, the article failed to summarize trends and publication gaps on the subject. Thus, authors should highlight the main contributions of their article to the academic literature, comparing it with previously published articles, in order to attract the reader's interest.

The work cites 30 articles, a relatively low number for a high-impact article. Considering the relevance of the theme and the impact of this journal, the Literature Review section should be expanded, presenting the newest algorithms applied in the analysis of quality information in world public health, such as:

https://doi.org/10.3390/healthcare10112147

Finally, the results and conclusions must be improved, explaining the main contributions of the paper to the scientific community and society.

Author Response

We thank the Reviewer for the comments and suggestions.

  1. Regarding the introduction section: we implemented this section by introducing and discussing new relevant references in order to better contextualize the readers in the immunotherapy field. The new section reads as follow:

(p 2, line 35-62):”The IMT role in the management of urological cancer patients is strongly corroborated by solid evidences and supported by the most recent European Association of Urology (EAU) guidelines, specifically for renal cell carcinoma (RCC) and for bladder cancer (BC) (8–19). Conversely, for upper tract urothelial carcinoma (UTUC), most of the evidence came from BC trials results, due to the low incidence of the disease (4,20–23). Even more uncertain is the role of IMT for prostate cancer (PCa), where the number of PCa patients eligible for IMT management is limited. Regardless of the urological tumor treated, the IMT approach has been previously considered a reasonable option for patients with metastatic disease (21). Thanks to the promising published results, the scientific scenario is changing over the years, including a setting of patients with an earlier stage disease. For instance, the PURE-01 study assessed the safety of Pembrolizumab in neoadjuvant setting in muscle invasive bladder cancer (MIBC) obtaining an a higher rate of pT0 after surgery, relative to previous protocols which did not include IMT (24).

IMT represents a complex topic for both people with and without medical background. The spread of high-quality information on IMT trough social media might represent a potential instrument for educational purpose used by physicians and by general users who could be interested into the topic. Specifically, for the patients affected by IMT eligible disease, the access to information could increase their awareness in order to make more manageable the diagnostic-therapeutic steps (25–29)

The searching and spreading of medical information on Social Media (SoMe) has grown rapidly (30). Specifically, after COVID-19 outbreak, YouTube™ become one of the most widely used platform to provide and get information (31). More and more studies had revealed the low-quality of medical content uploaded on YouTube™, enlightening the potential risks and consequences of misinformation spreading to Internet users (32–35). To the best of our knowledge, we are the first to evaluate the quality of information on IMT in urological tumors, such as urothelial carcinoma (UC), RCC, and PCa, uploaded on YouTube™ during the last decades”.

  1. Regarding the comparison with previously published articles: to the best of our knowledge, we are the first to examine the YouTube video content on IMT. So a comparison cannot be done. However, we emphasised this concept in the new version of the manuscript:

(p 2, line 60-62):” To the best of our knowledge, we are the first to evaluate the quality of information on IMT in urological tumors, such as urothelial carcinoma (UC), RCC, and PCa, uploaded on YouTube™ during the last decades

(p 9, line 263-268): “To the best of our knowledge, we are the first to evaluate videos on IMT for urological cancers uploaded on YouTubeTM. We analyzed as many videos as possible collecting the first 30 videos for each keyword (9, for a total of 270 videos). Thus, we made the analysis as much representative as possible of what currently be uploaded on YouTubeTM on the topic. Despite these observations, several further studies are required to better understand and characterize the quality of Internet information on IMT.

  1. Regarding the references number: we included more than 20 new pertinent references in the new version of the manuscript: PMID:

  1. Regarding the improvements of results discussion and conclusions: we implemented the new version of the manuscript, which now read as follows:

Discussion section:

(pag 9, line 270-285): “Due to the complexity of the topic and the involved patient, Internet and particularly YouTubeTM may represent a crucial instrument to improve the communication between physicians and patients. Indeed, the videos could be an effective tool to improve patient awareness thanks to the visual provision of information that could overcome some health literacy barriers (54). Moreover, the YouTubeTMvideos should be uploaded to SoMe platforms to increase high-quality spread of information.

Conclusion section:

(pag 10, line 315-321): “Despite the multimedia content was evidence based, it had not an educational role for Internet users. Moreover, YouTubeTM videos contributed to spread misinformation specifically underestimating the role of IMT in a multimodality approach and omitting the results achieved from current clinical trials.

Round 2

Reviewer 2 Report

The authors were able to greatly increase the quality of the paper by incorporating the reviewers' recommendations. I suggest accepting the article for publication.